# Effect of Ultrasound Treatment on Barrier Changes of Polymers before and after Exposure to Food Simulants

**DOI:** 10.3390/polym14050990

**Published:** 2022-02-28

**Authors:** Mario Ščetar, Davor Daniloski, Mirela Tinjić, Mia Kurek, Kata Galić

**Affiliations:** 1Faculty of Food Technology and Biotechnology, University of Zagreb, Pierottijeva 6, 10000 Zagreb, Croatia; mscetar@pbf.hr (M.Š.); del.mirela@gmail.com (M.T.); mkurek@pbf.hr (M.K.); kgalic@pbf.hr (K.G.); 2Advanced Food Systems Research Unit, Institute for Sustainable Industries and Liveable Cities and College of Health and Biomedicine, Victoria University, Melbourne, VIC 8001, Australia; 3Food Chemistry and Technology Department, Teagasc Food Research Centre, Moorepark, Fermoy, P61 C996 Cork, Ireland

**Keywords:** linear low-density polyethylene, coated polypropylene, ultrasound, barrier properties, food–packaging interaction, overall migration

## Abstract

In this study, we investigated the impact of ultrasound treatment on barrier properties of linear low-density polyethylene (LLDPE) and acrylic/poly(vinylidene chloride) polypropylene (PPAcPVDC)-coated pouches intended for food packaging before and after exposure to food simulants. Packaging pouches were filled with two food simulants, namely ethanol (10% (*v*/*v*)) and acetic acid (3% (*w*/*v*)), in order to simulate food–packaging interaction and possible compound migration from packaging materials. Samples were subjected to an ultrasound water bath treatment for 5 min, 15 min, and 30 min at 60 °C (±2 °C) and with an amplitude of 100% as an equivalent to the heat-treatment conditions combined with an ultrasound effect. Furthermore, the effect of temperature on the polymer barrier (water vapour and oxygen permeability) properties was tested at 20 °C, 40 °C, and 60 °C. Results showed that PPAcPVDC possessed better properties of water vapour permeability and oxygen permeability properties to LLDPE. Statistical analyses showed a significant (*p* < 0.001) impact of ultrasound treatment on the overall migration value, regardless of the food simulant used.

## 1. Introduction

The main role of packaging is to protect foods from unfavourable external factors, including gases and vapour, and to preserve product characteristics (quality and microbial safety) during the shelf life [1,2]. Polyolefins (polyethylene and polypropylene) are known as the most commonly used food packaging materials due to various material properties (chemically inert, thermosealable, excellent gas and moisture barrier, etc.) [3].

Polypropylene (PP) is a polymer available at low cost that has excellent thermal stability, optical transparency, and barrier properties [4]. Linear low-density polyethylene (LLDPE) provides a high barrier to gases and has high processing adaptability to various shapes of food packaging [5,6,7]. It is often used for production of thinner plastic films. Its main limitation is high deformation after application of mechanical force (even at a room temperature) [7]. Compared to PP, LLDPE materials show lower gloss, greater haze, have smaller heat-sealing capacity, and, to a lesser extent, are appropriate for shrink films [8].

Ultrasound treatment (UST) is defined as treatment with acoustic waves with frequencies between 20 kHz and 100 kHz. UST is a non-ionizing, non-invasive, and non-polluting form of mechanical energy. It represents a novel method for use in food technology, with the ability to control and improve the preservation of treated food [9,10]. However, this non-invasive method might have an impact on the food packaging material if used as in-package treatment. Thus, knowledge on changes in barrier performance, mechanical properties, and risk of migration of unwanted substances from material to food must be seriously considered and studied before UST is applied in real-life scenarios. The impact of UST on the properties of packaging films, including bio-based and non-synthetically produced materials (polylactic acid, polyethylene furanoate, polybutylene succinate, polyhydroxyalkanoate, cellulose, starch, proteins, lipids, and waxes), as well as synthetically produced materials (PP, PE, polyvinyl chloride), indicates that this technology can be successfully utilised to enhance their barrier properties [4,11,12,13]. Recently, several studies reported positive effects of UST on food packaging materials (significant increase in film strength, elasticity, and hydrophobicity), as well as the effect of sonication on polymer degradation [14,15,16,17,18,19].

Various small compounds (monomers, additives, or unintended chemical compounds present during material production) could eventually migrate to packaged food, resulting in a dietary consumption of chemicals that may be harmful to human health [20]. Because different novel techniques have been used for packaged food preservation, such as UST, it is of great importance to evaluate the overall migration value before launching novel products on the market. To our knowledge, there is no such literature data dealing with the impact of ultrasound on substance migration in food simulants (FSs).

The present study focuses on the determination of barrier performance (water vapour and oxygen permeability) of two commercially available food packaging films, namely linear low-density polyethylene (LLDPE) and polypropylene coated with acrylic/poly (vinylidene chloride) (PPAcPVDC), both treated with a UST. In addition, we determined the impact of the FS (ethanol (EtOH) and acetic acid (HAc)) on the packaging film barrier properties during UST, as well as the overall migration (OM) in FSs, as influenced by UST.

## 2. Materials and Methods

### 2.1. Packaging Materials and Chemicals

In this study, we used linear low-density polyethylene-LLDPE (Simplast, Monta, Italia; Sima Flex-L2G; 50 μm) and polypropylene coated with acrylic/poly(vinylidene chloride)-PPAcPVDC (ExxonMobil, Machelen, Belgium; MB 777; 32 μm), without any graphic treatment. Ethanol (EtOH, 10% (*v*/*v*)) (Gram-Mol, Zagreb, Croatia) and acetic acid (HAc, 3% (*w*/*v*)) (J. T. Baker, Phillipsburg, NJ, USA) were used for preparation of FSs.

### 2.2. Sample Preparation

LLDPE and PPAcPVDC films were cut to sheets of 24 cm × 18 cm and sealed into packaging pouches. The active surfaces exposed to the action of FSs was 352 cm^2^. The pouches were filled with 100 mL of two different FS solutions: aqueous ethanol solution at 10% (*v*/*v*) and aqueous acetic acid at 3% (*w*/*v*), representing hydrophilic and acidic foodstuff, respectively. The samples were coded according to FS type and UST (Appendix A, Appendix A).

### 2.3. Ultrasound Treatment

Ultrasound treatment was performed with an ultrasonic water bath (Sonorex Digiplus DL 255 H, Bandelin, Heinrichstrabe, Berlin, Germany) operated at 35 kHz. The maximum ultrasonic power of the system was used (640 W adjusted to an amplitude of 100%). All samples were treated at 60 °C to simulate the in-package heat treatment for 5, 15, and 30 min (Appendix A). After the treatment, FSs were first poured out of the pouches, and then the excess of food simulant was carefully removed using filter paper. Then, films were used for further barrier and migration measurements. Depending on the packaging material, treatment conditions (UST), food simulant (EtOH or HAc), and the heating time (5, 15, or 30 min), the samples were denoted as LLDPE (control); LLDPE: EtOH 5, 15, or 30; LLDPE: HAc 5, 15, or 30; PPAcPVDC (control); PPAcPVDC: EtOH 5, 15, or 30; and PPAcPVDC: HAc 5, 15, or 30 (Table 1, Table 2, Table 3 and Table 4; Appendix A).

LLDPE—linear low-density polyethylene; PPAcPVDC—polypropylene coated with acrylic/poly(vinylidene chloride); control—treatment time equal to 0 min; EtOH—10% (*v*/*v*) ethanol; Hac—3% (*w*/*v*) acetic acid. Mean values within a column that do not share a common superscript letter are significantly different (*p* ≤ 0.05).

PPAcPVDC—polypropylene coated with acrylic/poly(vinylidene chloride; control—treatment time equal to 0 min; EtOH—10% (*v*/*v*) ethanol; Hac—3% (*w*/*v*) acetic acid. Mean values within a column that do not share a common superscript letter are significantly different (*p* ≤ 0.05).

LLDPE—linear low-density polyethylene; control—treatment time equal to 0 min; EtOH—10% (*v*/*v*) ethanol; Hac—3% (*w*/*v*) acetic acid. Mean values within a column that do not share a common superscript letter are significantly different (*p* ≤ 0.05).

LLDPE—linear low-density polyethylene; PPAcPVDC—polypropylene coated with acrylic/poly(vinylidene chloride; control—treatment time equal to 0 min; EtOH—10% (*v*/*v*) ethanol; Hac—3% (*w*/*v*) acetic acid; nd—not determined. Mean values within a column that do not share a common superscript letter are significantly different (*p* ≤ 0.05).

### 2.4. Film Characterisation

#### 2.4.1. Film Thickness

The film thickness was determined with a digital micrometer (accuracy of 0.001 mm; Digimet, HP, Helios Preisser, Gammertingen, Germany). The average value at five positions per type of sample was used in all calculations.

#### 2.4.2. Water Vapour Permeability

The water vapour permeability (*WVP*) of films was determined under controlled conditions according to the gravimetric standard method [21] adapted by Basiak, Debeaufort, and Lenart (2016), with minor modifications, i.e., a relative humidity (RH) differential of 30% to 100% RH and a temperature of 25 ± 1 °C (Memmert climatic ventilated chamber HPP110, Memmert, Büchenbach, Germany) [22]. Samples were equilibrated for 72 h (25 ± 1 °C, RH 70%) before measurement. During the measurement, all samples were weighed twice a day. The *WVP* (g·m^−1^·s^−1^·Pa^−1^) was calculated from the change in the cell weight versus time at the steady state, using the following Equation (1):(1)$$WVP=\frac{\Delta m}{\Delta t\;\cdot \;A\;\cdot \;\Delta p}\cdot x$$
where Δ*m*/Δ*t* is the weight loss per unit of time (g·s^−1^), *A* is the film area exposed to the moisture transfer (9.08 × 10^−4^·m^2^), *x* is the film thickness (m), and Δ*p* is the water vapour pressure difference between the two sides of the film (Pa). All measurements were performed in triplicate.

#### 2.4.3. Oxygen Permeability

The gas permeance (*q*: cm^3^ ∙ m^−2^ ∙ d^−1^ ∙ bar^−1^) was determined according to the International Organization for Standardization (ISO) [23] on a GDP-C-type permeability-testing instrument (Brugger Feinmechanik GmbH, München, Germany). The experiments were conducted at 20 °C, 40 °C, and 60 °C and controlled by an external water bath (Haake F3 with Waterbath K, Karlsruhe, Germany). Measurement was performed using the selected method that permits readings of permeability (*P*: cm^3^·cm^−1^·s^−1^·bar^−1^), solubility (*S*: cm^3^·cm^−3^·bar^−1^), and diffusion (*D*: cm^2^∙s^−1^) coefficients, calculated from the time lag (*tL*: s) value at the provided sample thickness [24]. All coefficients were obtained in triplicate with good reproducibility, and the mean was calculated. We also obtained the activation energy for oxygen diffusion (*E_D_*), permeability (*E_P_*), and heat of sorption (*E_S_*) through untreated and UST-treated LLDPE and PPAcPVDC films.

#### 2.4.4. Overall Migration

The overall migration (OM) values were determined to check the maximal quantity of low-molecular-weight compounds that had migrated from the packaging materials to the FSs. Ethanol (10% (*v*/*v*)) and aqueous acetic acid (3% (*w*/*v*)) were used as FSs for all measurements. Measurements were performed with a migration cell (MigraCell^®^; FABES Forschungs-GmbH, Munich, Germany) and either following (1) the immersion method (for untreated and control samples with UST only) or (2) the article filling method (for pouches filled with FSs and UST-treated samples) according to the modified EN 1186-1 and EN 1186-5 standards [25,26], respectively. The surface (A/dm^2^) of the sample was measured prior to experiments. Migration cells (for method 1) or pouches (for method 2) were kept for 10 days at 40 °C. Afterwards, FS solution from the upper part of the migration cell (for method 1) and from the pouches (for method 2) was decanted in a previously weighed glass cell and evaporated at high temperatures (>300 °C). All measurements were performed in triplicate. The overall migration (OM) was calculated according to the following equation:(2)$$\mathrm{OM}=\frac{m_{2}-m_{1}}{A}\;\;\;\;\;\;\;\;(\mathrm{mg}\;\cdot \;\mathrm{dm}{-}^{2})$$
where *m*_1_ is the mass of the empty glass cell before evaporation (mg), *m*_2_ is the mass of the glass cell after evaporation (mg), and *A* is the surface of the test specimen intended to come into contact with the simulant (dm^2^). All measurements were performed in triplicate.

### 2.5. Data Analysis

Statistical analysis was performed in Minitab using analysis of variance (ANOVA) (Version 19; Minitab, PA, USA), with packaging material (Pack), ultrasound treatment (UST), food simulant (FS), and temperature (Temp) as fixed factors. Tukey’s test was used as the post hoc test, and *p* < 0.05 was considered significant. The permeability data (*S*, *D*, and *P*) were evaluated and reported with Origin software (Origin Pro 2021, v. 95E, OriginLab Corporation, Northampton, MA, USA). All experiments were carried out at least in triplicate, and results were reported as the means and standard errors of differences of the means of these measurements.

## 3. Results and Discussion

### 3.1. Water Vapour Permeability

The PPAcPVDC (control) (Table 1) showed better barrier properties for water vapour (1.55 × 10^−13^ g·m^−1^·s^−1^·Pa^−1^) compared to the LLDPE (control) (3.51 × 10^−13^ g·m^−1^·s^−1^·Pa^−1^) (*p* < 0.001). When the UST time was increased from 5 min to 30 min (both food simulants (FSs) used), the *WVP* of PPAcPVDC significantly decreased (*p* < 0.001, Appendix A). This could be due to the change in the structure of the packaging material under the action of both UST and FSs, thus reducing the rate of water vapour passage through the film. Wang et al. (2014) also showed that *WVP* of ultrasound/microwave treated soybean protein isolate films (addition of carboxymethyl cellulose, oleic acid, and stearic acid: *WVP* = 0.1336 × 10^12^ g·cm^−1^·s^−1^·Pa^−1^) was decreased compared to the control (only carboxymethyl cellulose added: *WVP* = 0.4525 × 10^12^ g·cm^−1^·s^−1^·Pa^−1^) [27]. Ščetar et al. (2019) conducted a microscopic analysis of PP packaging films and observed large crystal agglomerates on the surface of the film, which (6 min and 100% amplitude) became more scattered under the influence of UST [15]. Thus, the presence of crystal structures makes the films almost impermeable to water vapour. This behaviour can be considered a desirable material property for hydrophilic foodstuff [28,29].

### 3.2. Oxygen Permeability Parameters

Results of the oxygen permeability parameters for PPAcPVDC and LLDPE are given in Table 2 and Table 3, respectively. In general, when comparing these two materials, it can be observed that the permeability parameters were higher for LLDPE samples than for PPAcPVDC, as expected, due to the presence of acrylic/PVDC coating in PP film. Permeability values of LLDPE-treated samples (UST and FSs) were the highest at 40 °C (Table 3), whereas in PPAcPVDC-treated samples (UST and FSs), permeability increased as the temperature increased (*p* < 0.001) (Table 2). The exception was found for PPAcPVDC: EtOH 30 min at 40 °C. There was no particular explanation for this behaviour, so it could be a combined effect of UST and FS on the linear polymer structure, which was not the case for PPAcPVDC due to the presence of a coating. Such effect should be further investigated. It is generally claimed that if the temperature rises, the permeability coefficients will also increase [4,30,31]. This is mainly due to the increased energy and activity of polymer chains, which facilitate the movement of polymer macromolecules, creating a gap between polymer units, resulting in an increased gas permeability [32,33,34]. All samples treated with UST had higher permeability values compared to the control samples at a given temperature; however, a particular conclusion regarding the treatment time was not revealed. The influence of high-power ultrasound on the change in chemical and mechanical properties of polymer films has been reported in previous studies [14,15,35]. Accordingly, Klepac, Ščetar, Baranović, Galić, and Valić (2014) showed that gamma radiation with ^60^Co γ-source and a dose of 200 kGy on PP_cast_ and LLDPE, commonly used as a non-thermal preservation method, reduced permeability coefficients (*S*, *D*, and *P*) for both tested polymers [36]. Namely, after the treatment, *P* decreased by about 8% for LLDPE and 19% for PP_cast_ film [36].

As already mentioned, the better barrier characteristics of PPAcPVDC compared to LLDPE were mostly attributed to the presence of an AcPVDC coating. Similarly, Daniloski et al. (2019) showed that PPAcPVDC film was a better barrier to oxygen compared to biaxially oriented coextruded PP [31]. Moreover, Leterrier (2003) showed that PVDC coating had very low *P* (8.6 × 10^−3^ cm·m^−2^·d^−1^·bar^−1^), which was responsible for the remarkably increased barrier properties of biaxially oriented PP films [37].

In both control samples, the oxygen solubility coefficient (*S*) (Table 2 and Table 3) increased with increasing temperature (40 °C to 60 °C, *p* < 0.001). A similar trend was observed for most of the treated (UST and FSs) PPAcPVDC samples, except for PPAcPVDC: HAc 15 min, and no specific behaviour could be seen for the treated LLDPE samples. Mrkić, Galić, Ivanković, Hamin, and Ciković (2006) and Daniloski et al. (2019) observed a significant increase in the *S* of gases at higher temperatures (above 50 °C) in PE and monoaxially and biaxially oriented PP films [31,38]. According to Ščetar et al. (2019), it is possible that the lower *S* (as obtained for PPAcPVDC samples) was due to the larger volume fraction of the crystal region in contrast to the LLDPE samples [15]. In theory, the crystalline regions in polymers are impermeable, with practically no sorption of gas molecules, whereas the volume fraction of the amorphous parts is responsible for the gas transport within their structure [39].

In the LLDPE (control) samples, *D* values slightly increased with increased temperature. In LLDPE samples treated with UST and EtOH (Table 3), there was an important impact of temperature (*p* < 0.001, Appendix A), whereas no significant impact could be seen for LLDPE samples treated with UST and HAc (*p* > 0.05); the exception was LLDPE: HAc 30 min. In PPAcPVDC (control) samples (Table 2), there was a significant increase in *D* at 60 °C, whereas PPAcPVDC samples treated with UST and EtOH showed a decrease in *D* with increasing temperature. Lower *D* compared to *S* in all samples confirms that permeation of oxygen through tested polymers was under the diffusivity control [40].

The Arrhenius plots of *S*, *D*, and *P* for both PPAcPVDC and LLDPE films are presented in Figure 1, Figure 2 and Figure 3, respectively. All PPAcPVDC (control) films followed the Arrhenius relationship for *S* and *D*, with good correlation (for *S* data: PPAcPVDC= 0.9913, *p* < 0.001; LLDPE = 0.9998, *p* < 0.001; and for *D* data: PPAcPVDC = 0.7786, *p* < 0.001; LLDPE = 0.9533, *p* < 0.001). Similarly, Daniloski et al., (2019) found a good correlation for *D* values of biaxially oriented PPAcPVDC (0.82, *p* < 0.001) [31]. In the current study, the untreated samples (PPAcPVDC (control) and LLDPE (control)) did follow the Arrhenius relationship for *P*. Hong and Krochta (2006) showed a good fit of the Arrhenius plot for *P* of PP and PE films (PP = 0.998; and PE = 0.999) [41]. According to the statistical analysis, the correlation between the packaging material, temperature, and UST significantly influenced all permeability parameters (*p* < 0.001). *S*, *D*, and *p* values were not significantly affected (*p* > 0.05) by UST.

With the activation energy (*Ea*) for *P*, *D*, and *S,* the rate and affinity of transferring molecules can be predicted under various conditions. *Ea* is defined as the energy required to start the diffusion of a particular gas through the packaging material [42]. The activation energies calculated from the obtained results are presented in Appendix A. The *Ea* values of all parameters in the LLDPE (control) samples were lower than in the PPAcPVDC (control) samples. PPAcPVDC was shown to be a better barrier against oxygen; the *Ea* of PPAcPVDC was higher than that of LLDPE. After UST in contact with FSs, all *Ea* values for *P* increased compared to the control samples. *Ea* (*D*) and *Ea* (*S*) were lower for UST at 5 min, with no particular behaviour in other samples. An ample knowledge of these parameters is demanded for estimating the efficiency of packaging material and the quality properties of the packed food during storage, including the distribution chain. Fluctuations in temperature might lead to changes in material permeability, affecting the transfer of gases through the package, resulting in limited shelf life of the packed food item [43].

### 3.3. Overall Migration

Overall migration (OM) tests are performed to determine the overall quantity of chemicals that may migrate from polymer films into food (or a simulant). Namely, migration could be affected by numerous factors, such as packaging materials, food nature, temperature, characteristics of migrating additives, and the number of potential migrants in packaging materials, to name a few [44]. Polyolefin materials, specifically PP and PE, are composed by the polymerisation of hydrocarbons (containing only CH_2_ groups) and possess low wettability due to their hydrophobic properties. The passage of migrants through these materials in normal conditions (without any treatment, including UV-irradiation, γ-sterilisation, thermal, plasma, or UST treatments) is very unlikely [45]. The OM results of LLDPE and PPAcPVDC with or without UST are given in Table 4. The two control samples had the lowest OM values, followed by ultrasound-treated samples. The migration from LLDPE (control) was higher in both simulants compared to PPAcPVDC (control) (*p* < 0.05). Nevertheless, the above-mentioned treatments might either decrease the hydrophobicity of the polymer or introduce C=O or –OH groups that are more prone to degradation. Once treated, it was observed that PP was more prone to degradation in comparison to PE [46]. Similarly, Ščetar et al. (2017) observed an increase in the hydrophilic properties of biaxially orientated PPAcPVDC after performing high-power UST for 6 min and 100% amplitude [4]. These instances are in line with the present study’s results, which indicated that with UST, the OM of the packaging materials increased; however, OM was higher in PPAcPVDC than in LLDPE (Table 4). The highest OM values were determined in samples where food simulants were packed in pouches and then treated with UST (*p* < 0.05). This indicates that there is an impact of the packed simulant or foodstuff on the surface of the material during UST (*p* < 0.05). Several authors have explained that a number of factors could affect these adjustments [4,47,48]. Namely, physical or chemical changes of the surface layer of the film due to cavitation may lead to microscopic defects on the material’s surface and the rapid growth and explosive collapse of microscopic bubbles as the alternate compression and rarefaction phases of the sound wave pass through the liquid once the UST has been obtained. Eventually, the energy released in the created air bubbles might burst the film surface, leading to changes in surface morphology, including changes in roughness, texture, and heterogeneity [4,47,48].

Additionally, for both treatment methods, only UST on raw material and pouches filled with FS showed higher OM values for HAc in comparison with EtOH-exposed samples (not comparing both packaging materials, Table 4). In this regard, the observed differences in OM can be attributed to the affinity of both PPAcPVDC and LLDPE polymers and FSs. It can also be observed from the results that the affinities between PPAcPVDC and HAc, as well as LLDPE and Hac, were higher for almost 35% and 51%, respectively, compared to the affinities between PPAcPVDC and EtOH and LLDPE and EtOH. As a result of this phenomenon, the penetration of prior FS into the polymer matrices during migration was significantly higher compared to the latter FS (*p* < 0.05). Hafttananian, Zabihzadeh Khajavi, Farhoodi, Jahanbin, and Ebrahimi Pure (2021) presented that LDPE possessed more affinity to HAc than distilled water, and as such, a greater penetration of HAc was observed in the polymer matrix [49]. Interestingly, after the UST, for both FSs, LLDPE showed less affinity, which led to less migration of both simulants through this material compared to PPAcPVDC (*p* < 0.05). A similar observation was made by Liu Jing-Min et al. (2020), who investigated the migration of HAc through PE wrap films. The authors observed a poor affinity of the PE wrap film for HAc, which caused a small swelling effect of the film and, consequently, less migration of that FS through the packaging material [50].

The statements in the above paragraphs might indicate that tested materials in combination with UST might not be acceptable for acidic (OM results obtained for HAc) and hydrophilic (OM results obtained for EtOH) foodstuff. Correspondingly, the OM values should be less than the legally permitted values of 10 mg·dm^−2^, as previously stated elsewhere in the literature [51]. Notably, the treatment conditions in this study were extremely demanding, with a high temperature (60 °C) and a long period of exposure (30 min with the highest UST amplitude). From the obtained results, it can be also seen that all three fixed parameters considered (packaging material (Pack), ultrasound treatment (UST), and food simulant (FS)), were shown to be significant factors influencing the OM (*p* < 0.001, Appendix A). Based on the migration results during UST, LLDPE films were more stable than PPAcPVDC; however, knowledge on the extent of migration of substances from materials is important for food packaging, as it might present a health risk for consumers.

## 4. Conclusions and Future Recommendation

LLDPE and PPAcPVDC, commonly used commercial food packaging materials, were treated with an ultrasound while in contact with two FSs representing hydrophilic and acidic foodstuff. Treated (UST and FSs) LLDPE and PPAcPVDC showed different barrier properties for water vapour and oxygen under the tested conditions. PPAcPVDC was shown to be a better gas barrier in contrast to LLDPE. Temperature was presented as a statistically significant factor for permeability properties, leading to a remarkable increase in the permeability properties of the packaging material samples with temperature increase (from 20 °C to 60 °C, *p* < 0.001). The diffusion process controlled oxygen permeation through both materials. In LLDPE, the lowest *WVP* was reached for the control sample, and the highest for the sample treated with UST and HAc for a period of 30 min. On the contrary, in PPAcPVDC, the treated samples (UST and FSs) were less permeable to water vapour than the control samples. The OM was shown to be significantly higher when samples were treated with UST and in contact with FSs.

Results obtained in this study indicate that UST had an impact on the barrier properties of the tested materials. As novel processing methods (such as ultrasound) are achieve commercial importance, it is envisaged to study the physicochemical and surface properties of the tested materials. Therefore, further studies are still needed to extend our results to real foodstuffs because different food products have specific packaging requirements based on their permeability characteristics and storage conditions.

## Figures and Tables

**Figure 1 polymers-14-00990-f001:**
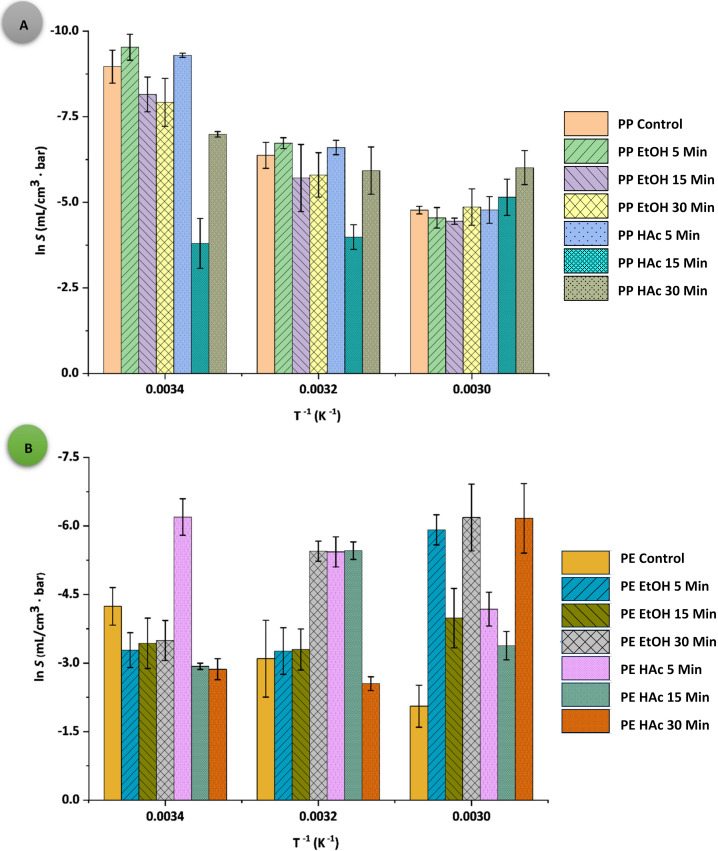
Arrhenius plots of gas solubility (ln *S*) as a function of temperature (*T^-1^*); (**A**) PPAcPVDC and (**B**) LLDPE samples.

**Figure 2 polymers-14-00990-f002:**
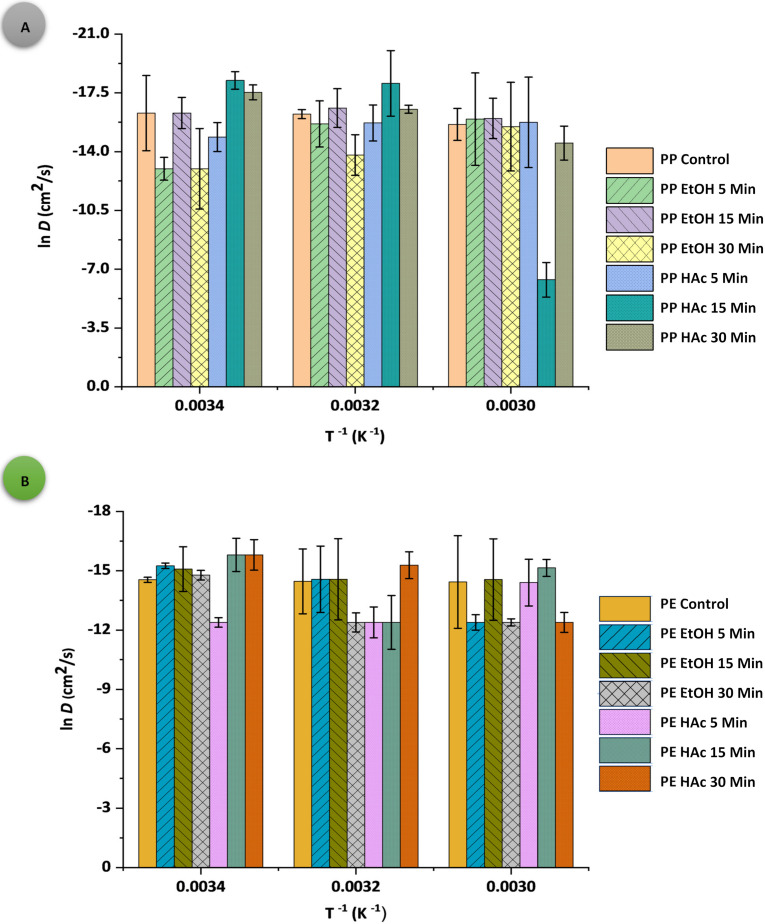
Arrhenius plots of gas diffusion (ln *D*) as a function of temperature (*T*^−1^); (**A**) PPAcPVDC and (**B**) LLDPE samples.

**Figure 3 polymers-14-00990-f003:**
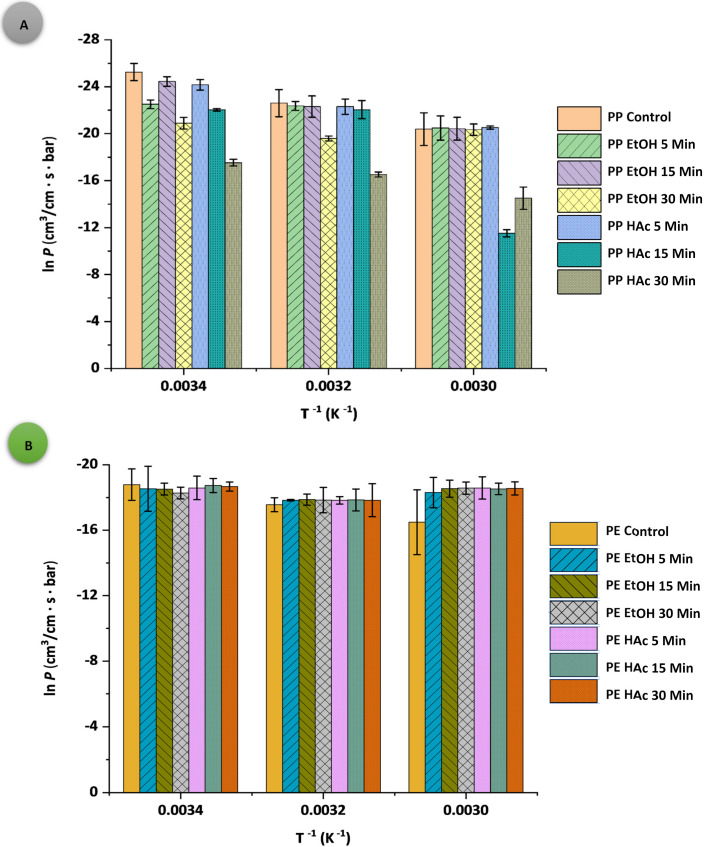
Arrhenius plots of gas permeability (ln *P*) as a function of temperature (*T*^−1^); (**A**) PPAcPVDC and (**B**) LLDPE samples.

**Table 1 polymers-14-00990-t001:** Thickness values and water vapour permeability (*WVP*) of LLDPE and PPAcPVDC before (control) and after ultrasound treatment (UST: 5 min, 15 min, or 30 min) at 25 ± 1 °C and 70% RH.

Sample: Treatment Conditions	Thickness (μm)	*WVP* × 10^−13^ (g·m^−1^·s^−1^·Pa^−1^)
LLDPE (control)	51 ± 1.54 ^ab^	3.51 ± 0.32 ^c^
LLDPE: EtOH 5	50 ± 1.01 ^abc^	4.03 ± 0.17 ^bc^
LLDPE: EtOH 15	52 ± 0.24 ^a^	4.10 ± 0.20 ^bc^
LLDPE: EtOH 30	50 ± 0.73 ^abc^	4.07 ± 0.12 ^bc^
LLDPE: HAc 5	48 ± 1.01 ^bc^	10.3 ± 0.53 ^a^
LLDPE: HAc 15	48 ± 0.98 ^bc^	3.95 ± 0.08 ^bc^
LLDPE: HAc 30	51 ± 0.68 ^ab^	4.28 ± 0.27 ^b^
PPAcPVDC (control)	34 ± 0.51 ^d^	1.55 ± 0.06 ^d^
PPAcPVDC: EtOH 5	30 ± 0.98 ^e^	1.05 ± 0.04 ^de^
PPAcPVDC: EtOH 15	29 ± 0.45 ^e^	1.17 ± 0.07 ^de^
PPAcPVDC: EtOH 30	32 ± 0.14 ^de^	1.08 ± 0.05 ^de^
PPAcPVDC: HAc 5	30 ± 0.29 ^e^	0.99 ± 0.09 ^de^
PPAcPVDC: HAc 15	30 ± 1.26 ^e^	0.83 ± 0.03 ^e^
PPAcPVDC: HAc 30	30 ± 0.53 ^e^	0.81 ± 0.01 ^e^

Different superscripts (a–e) within a column indicate significant differences among samples (*p* ≤ 0.05).

**Table 2 polymers-14-00990-t002:** Oxygen permeability data (expressed as diffusion *D*, solubility *S*, and permeability *P* coefficient and permeance *q*), for PPAcPVDC film at different measuring temperatures (20 °C, 40 °C and 60 °C) and different ultrasound treatment times (0 min, 5 min, 15 min, and 30 min).

PPAcPVDC					
Sample: Treatment Conditions	*t* (°C)	*D* × 10^−11^ (cm^2^·s^−1^)	*S* × 10^−5^ (mL·cm^−3^∙bar^−1^)	*P* × 10^−6^ (cm^3^·cm^−1^·s^−1^·bar^−1^)	*q* (cm^3^·m^−2^·d^−1^·bar^−1^)
PPAcPVDC (control)	20	0.84 ± 1.61 ^j^	0.13 ± 0.04 ^b^	1.07 ± 0.94 ^t^	2.91 ± 0.17 ^a^
	40	0.89 ± 0.01 ^j^	1.71 ± 0.12 ^b^	15.25 ± 1.50 ^p^	41.30 ± 3.84 ^a^
	60	1.65 ± 1.23 ^g^	8.48 ± 0.24 ^a^	139.92 ± 9.76 ^c^	377.00 ± 8.99 ^a^
PPAcPVDC: EtOH 5	20	23.10 ± 1.53 ^a^	0.07 ± 0.02 ^b^	16.74 ± 1.54 ^o^	28.90 ± 3.68 ^a^
	40	1.60 ± 3.20 ^g^	1.20 ± 0.79 ^b^	19.20 ± 1.26 ^m^	52.00 ± 2.95 ^a^
	60	1.20 ± 1.08 ^i^	10.60 ± 0.31 ^b^	127.20 ± 2.28 ^e^	342.00 ± 16.00 ^a^
PPAcPVDC: EtOH 15	20	0.83 ± 0.01 ^j^	0.29 ± 0.02 ^b^	2.41 ± 0.65 ^r^	6.53 ± 0.26 ^a^
	40	0.62 ± 0.03 ^k^	3.31 ± 0.13 ^b^	20.55 ± 1.24 ^l^	55.50 ± 2.76 ^a^
	60	1.15 ± 0.31 ^i^	11.70 ± 0.51 ^b^	134.55 ± 9.61 ^d^	364.00 ± 3.29 ^a^
PPAcPVDC: EtOH 30	20	23.10 ± 1.49 ^a^	0.36 ± 0.01 ^b^	84.08 0.84 ^i^	145.00 ± 4.07 ^a^
	40	10.20 ± 4.45 ^c^	3.03 ± 0.01 ^b^	309.06 ± 1.06 ^a^	836.00 ± 13.82 ^a^
	60	1.88 ± 0.23 ^f^	7.75 ± 0.15 ^b^	145.70 ± 5.09 ^b^	393.00 ± 1.37 ^a^
PPAcPVDC: HAc 5	20	3.48 ± 0.79 ^e^	0.10 ± 0.02 ^b^	3.20 ± 0.24 ^q^	8.65 ± 0.12 ^a^
	40	1.51 ± 0.37 ^h^	1.40 ± 0.02 ^b^	20.53 ± 4.22 ^l^	55.30 ± 0.86 ^a^
	60	1.45 ± 0.17 ^h^	8.42 ± 0.05 ^b^	122.09 ± 0.96 ^g^	331.00 ± 0.89 ^a^
PPAcPVDC: HAc 15	20	0.12 ± 0.01 ^m^	22.40 ± 2.24 ^b^	26.88 ± 0.11 ^j^	72.50 ± 1.61 ^a^
	40	0.14 ± 0.04 ^m^	18.60 ± 1.60 ^b^	26.59 ± 0.72 ^k^	72.00 ± 1.33 ^a^
	60	17.10 ± 4.70 ^b^	5.81 ± 0.13 ^b^	99.35 ± 0.52 ^h^	301.33 ± 17.15 ^a^
PPAcPVDC: HAc 30	20	0.24 ± 0.03 ^l^	0.92 ± 0.03 ^b^	2.25 ± 0.53 ^s^	6.10 ± 0.41 ^a^
	40	0.66 ± 0.08 ^k^	2.67 ± 0.27 ^b^	17.80 ± 0.86 ^n^	48.20 ± 0.20 ^a^
	60	5.00 ± 1.34 ^d^	2.45 ± 0.28 ^b^	122.50 ± 1.64 ^f^	330.00 ± 1.95 ^a^

Different superscripts (a–t) within a column indicate significant differences among samples (*p* ≤ 0.05).

**Table 3 polymers-14-00990-t003:** Oxygen permeability data (expressed as diffusion *D*, solubility *S*, and permeability *P* coefficient and permeance *q*), for LLDPE film at different measuring temperatures (20 °C, 40 °C and 60 °C) and different ultrasound treatment times (0 min, 5 min, 15 min, and 30 min).

LLDPE					
Sample: Treatment Conditions	*t* (°C)	*D* × 10^−11^ (cm^2^·s^−1^)	*S* × 10^−5^ (mL·cm^−3^·bar^−1^)	*P* × 10^−6^ (cm^3^·cm^−1^·s^−1^·bar^−1^)	*q* (cm^3^·m^−2^·d^−1^·bar^−1^)
LLDPE (control)	20	4.83 ± 1.94 ^e^	14.40 ± 0.01 ^j^	695.52 ± 2.57 ^s^	1200.00 ± 6.33 ^o^
	40	5.24 ± 2.04 ^d^	45.20 ± 0.45 ^e^	2368.48 ± 3.97 ^b^	4100.00 ± 12.62 ^a^
	60	5.40 ± 1.17 ^c^	128.00 ± 5.03 ^a^	6912.00 ± 7.12 ^a^	1100.00 ± 1.17 ^p^
LLDPE: EtOH 5	20	2.39 ± 0.11 ^i^	37.50 ± 9.89 ^f^	896.25 ± 11.97 ^l^	1540.00 ± 2.64 ^ij^
	40	4.71 ± 1.15 ^e^	38.30 ± 2.21 ^f^	1803.93 ± 7.86 ^d^	3110.00 ± 1.55 ^c^
	60	41.70 ± 8.45 ^a^	2.70 ± 0.70 ^klm^	1125.90 ± 10.57 ^i^	1940.00 ± 3.30 ^g^
LLDPE: EtOH 15	20	2.82 ± 0.43 ^g^	32.30 ± 1.94 ^gh^	910.86 ± 2.09 ^j^	1570.00 ± 7.09 ^h^
	40	4.71 ± 0.97 ^e^	37.00 ± 0.66 ^f^	1742.70 ± 9.71 ^g^	3010.00 ± 1.63 ^e^
	60	4.77 ± 1.95 ^e^	18.60 ± 5.42 ^i^	887.22 ± 7.25 ^m^	1530.00 ± 4.21 ^jk^
LLDPE: EtOH 30	20	3.83 ± 0.11 ^f^	30.40 ± 0.25 ^h^	1164.32 ± 3.25 ^h^	2010.00 ± 1.70 ^f^
	40	41.70 ± 4.66 ^a^	4.30 ± 1.25 ^kl^	1793.10 ± 5.32 ^e^	3100.00 ± 0.89 ^c^
	60	41.70 ± 1.97 ^a^	2.06 ± 0.75 ^lm^	859.02 ± 3.89 ^o^	1480.00 ± 3.29 ^l^
LLDPE: HAc 5	20	41.70 ± 7.14 ^a^	2.04 ± 0.20 ^m^	850.68 ± 5.42 ^p^	1460.00 ± 39.85 ^l^
	40	41.70 ± 8.49 ^a^	4.37 ± 0.41 ^k^	1822.29 ± 9.09 ^c^	3150.00 ± 5.40 ^b^
	60	5.56 ± 1.71 ^b^	15.30 ± 1.82 ^j^	850.68 ± 23.03 ^p^	1470.00 ± 2.11 ^l^
LLDPE: HAc 15	20	1.37 ± 0.45 ^j^	53.50 ± 2.18 ^d^	732.95 ± 5.03 ^r^	1260.00 ± 1.31 ^m^
	40	41.70 ± 3.30 ^a^	4.25 ± 0.46 ^klm^	1772.25 ± 26.72 ^f^	3050.00 ± 1.13 ^d^
	60	2.65 ± 0.33 ^h^	34.00 ± 1.18 ^g^	901.00 ± 18.93 ^k^	1560.00 ± 2.00 ^hi^
LLDPE: HAc 30	20	1.37 ± 0.97 ^j^	57.00 ± 1.09 ^c^	780.90 ± 2.10 ^q^	1230.00 ± 1.11 ^n^
	40	2.31 ± 0.74 ^i^	78.10 ± 1.56 ^b^	1804.11 ± 11.78 ^d^	3120.00 ± 4.54 ^f^
	60	41.70 ± 2.24 ^a^	2.10 ± 0.89 ^lm^	875.70 ± 19.52 ^n^	1510.00 ± 0.40 ^k^

Different superscripts (a–s) within a column indicate significant differences among samples (*p* ≤ 0.05).

**Table 4 polymers-14-00990-t004:** Overall migration values for LLDPE and PPAcPVDC films before and after ultrasound treatment (30 min at 60 °C) with two food simulants (10 days at 40 °C).

Sample: Treatment Conditions	In Contact with Simulant during Ultrasound Treatment	Overall Migration (mg·dm^−2^) Food Simulant	
		HAc	EtOH
LLDPE (control)	No	2.27 ± 0.49 ^d^	2.04 ± 1.26 ^d^
LLDPE UST	No	4.66 ± 3.49 ^d^	2.43 ± 0.10 ^d^
LLDPE: HAc UST	Yes	147.73 ± 17.86 ^b^	nd
LLDPE: EtOH UST	Yes	nd	76.29 ± 5.36 ^c^
PPAcPVDC (control)	No	0.68 ± 0.10 ^d^	1.46 ± 0.10 ^d^
PPAcPVDC UST	No	5.92 ± 0.10 ^d^	7.77 ± 0.39 ^d^
PPAcPVDC: HAc UST	Yes	253.25 ± 19.48 ^a^	nd
PPAcPVDC: EtOH UST	Yes	nd	89.83 ± 3.60 ^c^

Different superscripts (a–d) within a column indicate significant differences among samples (*p* ≤ 0.05).

## Data Availability

Not applicable.

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
