# Peer review of "Effect of Ultrasound Treatment on Barrier Changes of Polymers before and after Exposure to Food Simulants"

_polymers, 2022, doi:10.3390/polym14050990_

Round 1
Reviewer 1 Report
In this manuscript the authors determined barrier performance of two commercially available food packaging films, namely linear low-density polyethylene (LLDPE) and polypropylene coated with 66 acrylic/poly(vinylidene chloride) (PPAcPVDC), both treated with an Ultrasound Treatment, as well as the impact of the Food Simulants (ethanol [EtOH] and acetic acid [HAc]) on the packaging film barrier properties during the UST. The work is quite interesting, and I believe, it will be a good resource for the future research. However, the readability of the manuscript needs to be improved, preferably carefully reviewing by a native English speaker. Also, the referencing throughout the manuscript is not appropriate. Some examples, not limited to, have been mentioned below. I recommend considering the manuscript for publication upon addressing following observations.
- (Line 79) “Materials were cut to sheets of 24 cm · 18 cm a….” which materials the authors are referring to?
- (Line 90) This sentence needs improvement “All samples were treated at 60 0C to simulate the heat treatment conditions (pasteurisation), during 5 min, 15 min, or 30 min (Supplementary Table 1).”
- (Line 90-92) Please check the sentence “After the treatment, FSs were poured out of the pouches, the excess of water was attentional removed using filter paper and films were used for further barrier and migration measurements.”
- (Line 100-101), The citation for the method adapted by Basiak, Debeaufort, and Lenart (2016) needs to be provided. The author included it in the reference section (reference 22) however, reference 22 is missing in the text.
- (Line 103-104) check the sentence “Samples were equilibrated during 72 h (25 ± 1 0C, RH 103 70 %) before all measurements.”
- (Line 117), Check the unit of permeability.
- The expressions A/dm2 (line 131) and m1/mg (line 135) are confusing.
- (Line 161) The sentence needs citation “Ščetar et al. (2019) conducted…”
- (Line 180) Rearrange the sentence “In all LLDPE treated samples….”
- (Line 188-190), Correct the sentence “This is mainly due to the increased energy and the activity of polymer chains, which facilities the movement of polymer macromolecules, creating the gap between polymer units resulting in an increased gas permeability.”
- Table 1,2, and 3, replacing sample/treatment with sample/treatment time will be more meaningful.
- Table 1,2, and 3, the term SEM is not clear. It should be elaborated in its first appearance. And, the significant of the values for ‘SEM” needs to be discussed in the text.
- Please check the referencing for the whole manuscript. The citation should be placed at the end of the sentence where the reference for the work is indicated. For example (Line 222-229 and throughout the manuscript), citations should be placed for the works when they are mentioned in the paragraph “Mrkić, Galić, Ivanković, Hamin, and Ciković (2006) and Daniloski et al. (2019) observed a significant increase in the S of gases at higher temperatures (above 50 0C) in PE and monoaxially and biaxially oriented PP films. According to Ščetar et al. (2019), it is possible that lower S (as obtained for PPAcPVDC samples) was due to the larger volume fraction of the crystal region in contrast to the LLDPE samples. In theory, the crystalline regions in polymers are impermeable with practically no sorption of gas molecules in them, while the volume fraction of the amorphous parts is responsible for the gas transport within their structure [39].”
- The symbols (P, S, D etc.) and abbreviations (e.g. SEM) should be defined in their first appearance in the manuscript. 15. (Line 290), citation is needed.
Author Response
We would like to thank Reviewer 1 for the detailed comments and suggestions for improving the manuscript and considering the manuscript for publication. After completion of the suggested edits, the revised manuscript has benefitted from an improvement in the overall presentation and clarity. The manuscript was checked for any grammatical errors by a native English speaker. Please find our responses to your comments below.

Reviewer 2 Report
The manuscript by D. Daniloski and coworkers describes the effects of ultrasound irradiation on the barrier provided by polymers for food packaging.
I think that the manuscript deserves attention for publication, however, major revisions are required, since the effects on several parameters appear quite randomly distributed, while no information is given on structural changes after ultrasond exposure.
The following issues should thus be addressed:
-The sentence “water vapour has a good durability” is not clear to me.
-The proceure “After the treatment, FSs were poured out of the pouches, the excess of water was attentional removed using filter paper and films were used for further barrier and migration measurements.” is not clear to me: which is the source of water?
-The value of WVP for LLDPE: HAc 5 appears not reliable without repetition/ error.
-Concerning the sentence “Accordingly, Klepac, Ščetar, Baranović, Galić, and Valić showed that gamma radiation […] reduced permeability coefficients”, it seems not in agreement with former observation of “higher permeability”. Moreover, why the result with gamma radiation are reported while no previous example with ultrasounds is included?
-How can PPAcPVDC: EtOH (20 min.) pass from 23 to 0.83 and then back to 23? Without an error the measurements appears not reliable
-P for PPAcPVDC: HAc15 is written in strange way (double punctuation)
-LLDPE: HAc displays scattered values for D, again, which are meaningless without an error.
-Concerning the sentence “Similarly, 31 found a good correlation” : is there a missing author?
-Concerning the sentence “Changes in the UST parameters were shown to be the only insignificant factor in the case of D” seesms not correct.
-Finally, since the films appear not stable for what concern OM, the films should be analyzed for structural and morphological point of view to understand the effects of the ultrasounds irradiation.
Author Response
We would like to thank Reviewer 2 for their valuable suggestion that surely improved our manuscript immensely. We have addressed the comments raised by the Reviewer and applied their suggestions and have provided an updated manuscript for your consideration. Please find our responses to your comments below.

Reviewer 3 Report
Dear Authors,
I read Your manuscript with great interest and consideration.
In my opinion it is well write and very well presented. The scientific content is high and the interest in the field is great.
In my opinion the manuscript could be accepted and published as it is.
Bests
Author Response
The authors are most thankful to Reviewer 3 for their comments, for the time taken and for the kind words.

Round 2
Reviewer 1 Report
I think, the manuscript is can be published in its present form. Please consider the following during proof reading.
- Line 89-90, the word 'during' can be replaced with 'for'. Please do the same in line 107.
- Please replace the word "facilities" with "facilitates" in the line 192.